# Coping strategies of women with obstetric fistula in the Northern Region of Ghana

Patience Akayila Asupulie[1]*, Samuel Adjorlolo[2], Gideon Puplampu[2]

1 Department of Nursing, Regional Hospital, Sunyani, Ghana, 2 School of Nursing and Midwifery, College of Health Sciences, University of Ghana-Legon, Accra, Ghana

* asupuliep@gmail.com

## Abstract

Obstetric fistula is a condition where women leak urine or faeces without control due to a hole created between where babies are born through and where urine or feces are stored. Incontinence of urine and feces is one major challenge facing women with obstetric fistula. The study assessed the coping strategies of women with obstetric fistula in the Northern Region of Ghana. The qualitative descriptive phenomenology study recruited 15 women with obstetric fistula through purposive sampling technique. Semi-structured interview guide was used to collect data through face-to-face interviews. The consents of participants were sought, and interviews were recorded, transcribed verbatim, coded and analyzed manually using Colaizi's thematic content analysis method. Using chamber pots as seats or chairs, wearing many clothes, use of rugs, pampers, or sanitary pads, avoiding bulky foods, use of perfumes or detergents, and frequently bathing and washing clothes were found coping strategies of women with obstetrics fistula. Also, control for incontinence, noise during walking and elimination of odor were found as justification for the use of the coping strategies. In addition, parents and siblings, health workers, neighbours and friends were the main source of support for living with incontinence while skin rashes, itching pain, fear, anxiety and grief were the physical and psychological effects of living with incontinence. Coping strategies including use of chamber pots as seats, wearing of many clothes, use of rugs and pampers, avoiding bulky foods, frequent bathing and washing were employed by women with obstetric fistula to manage incontinence of urine and feces. Support of parents, siblings, health workers, neighbours and friends played a significant role in their daily lives. Despite these efforts, women with obstetric fistula experience physical and psychological effects include skin rashes, pain, fear and grief highlighting the adverse impact of obstetric fistula on the general well-being of women.

**Data availability statement:** Study can be accessed on Zenodo via doi: 10.5281/zenodo.14918864.

**Funding:** Study was self funded and there was no external source of funding. Specifically, the study was fully funded by first Author, Patience Akayila Asupulie.

## Introduction

The effort to end obstetric fistula due to its negative influence on maternal health has received global attention in the past decade. Obstetric fistula is a condition in women where a hole is created between the birth canal and bladder and/or rectum due to prolonged or obstructed labour without access to timely medical treatment leading to uncontrollable leakage of urine and feces [1]. The condition leaves victims with body odour and unpleasant smells making it uncomfortable for others to come closer to them. It is estimated that about two million women are living with fistula and 50,000–100,000 women in the world develop obstetric fistula each year [2]. Africa alone accounts for 30,000–90,000 cases of women who develop obstetric fistula annually [3]. According to the Ghana Health Service, the number of obstetric fistula cases in Ghana every year ranges from 711 to 1,352 [4].

Many factors including lack of awareness, poor health seeking behaviour and lack of adequate health systems have been suggested as being responsible for obstetric fistula, but the occurrence of obstetric fistula is mainly due to obstructed and delayed labour [5]. According to Ryan [6], when obstructed labour is prolonged, the tissues between the vagina and the bladder or rectum are usually destroyed due to the pressure generated by the pushing of the baby's head against the woman's pelvis. The destruction leads to a hole being created between the vagina and the bladder or rectum resulting in leakage of urine and feces. Studies have shown that several factors predispose women to delayed and obstructed labour which consequently results in obstetric fistula if such women survive in childbirth. These factors include the age of women at the time of delivering the first child, lack of access to health care facilities, poor obstetrics care, poverty, illiteracy, and bad cultural practices such as female genital mutilation [7,8].

Women with obstetric fistula may directly suffer from various complications which include renal failure, vaginal stenosis or dyspareunia, infertility, and other infections. These can have a psychological, social, and economic impact on women. Psychologically, women with obstetric fistula feel ashamed of their conditions and develop mental stress out of constantly thinking about their situation [9]. In terms of their social lives, they are mostly scared to mingle with people to prevent stigma and as a result, suffer isolation. For instance, a study conducted in Malawi by Drew [10] to explore the quality of life of women with obstetric fistula, revealed that despite the improvement of their condition after surgical repair, they continued to suffer relationship problems. This affects their marriages because most of them are ignored by their husbands while others become divorcees because they cannot satisfy their partners. Moreover, victims of obstetric fistula cannot effectively engage in economic activities as most of them are ignored by employers and customers due to the bad smell around them [11]. These consequences of the condition are a burden not only to the victims but families, societies, and nations at large.

According to Jarvis [12], the issues affecting the needs and challenges of women with obstetric fistula are complex because they are embedded in the socio-cultural values of society. For example, women whose cultural beliefs suggest that obstetric fistula is punishment for adulterous persons will shy away from seeking medical care

[13]. While lack of funds for treatment is a challenge to some women with obstetric fistula, lack of access to well-resourced treatment centers prevents others from seeking surgical repair. Another challenge facing women with obstetric fistula in terms of treatment is limited health professionals and surgeons at the treatment centers.

Given the above challenges that prevent most women with obstetric fistula from getting treatment, the majority of them are forced to live with the condition in their communities, precipitating the question of how they cope with the condition. Existing studies have established that women with obstetric fistula generally adopt problem-focused coping strategies such as reducing fluid intake and avoiding sexual intercourse as well as emotion-focused coping strategies such as family support and having faith in God for healing [14]. Hygiene-focused strategies where women with obstetric fistula adopt frequent bathing and changing of clothes have also been identified as one way of coping with the condition [15]. Despite the existence of various studies on coping strategies for women with obstetric fistula, little is known about the justification for the use of these strategies and their effects on the women. This study was conducted to identify the coping strategies of women with obstetric fistula, the justification for adopting a particular strategy and the effects of the strategy on their physical and psychological health. The study highlights practical steps taken by women with obstetric fistula to manage incontinence, noise and odor. It also provides insights on the critical role of parents, siblings, health workers, friends and neighbours in supporting women with obstetric fistula while revealing the physical and psychological effects of the coping strategies such as skin rashes, itching, pain, fear, anxiety and grief despite these efforts.

## Methodology

### Ethics statement

The Ghana Health Service Ethics Review Committee granted approval certificate number GHS-ERC024/11/20 for the conduct of the study. A written consent was obtained from participants who took part in the study.

### Study design and setting

A descriptive phenomenology design was used to explore the coping strategies of women with obstetric fistula. This design was chosen because it offered the researchers the opportunity to explore the lived experiences of participants without interpreting them.

The study was conducted in the Northern Region of Ghana. It was the largest region in Ghana with a land surface area of 70,384 square kilometres (30% of the total land surface of Ghana) until the North East and the Savannah regions were carved out in 2018. Tamale is the regional capital with many public and private health facilities including a Fistula Centre for the treatment of women with obstetric fistula. The Tamale Metropolis was chosen for the study because it has high prevalence of obstetric fistula due to cultural practices such as Female Genital Mutilation in the region [7]. The setting also has relatively well-developed healthcare infrastructure including a Fistula Centre that cater for women with obstetric fistula. This facilitated access to study participants, enabling collection of rich data on coping strategies and experiences of women with obstetric fistula.

### Study population, sampling technique, and sample size

Women with obstetric fistula in the Northern Region of Ghana were the target population of the study. The inclusion criteria of the study involved women who were diagnosed with obstetric fistula at the Fistula Centre, aged 18 years old or more, and fluent in English, Dagbani, or Twi languages. Purposive sampling was used to select 15 participants for the study. Women with obstetric fistula who did not understand English, Dagbani or Twi and those who could not be contacted after the interviews for follow-up were excluded in the study. The sample size was determined based on data saturation. As the data was collected, each transcript was constantly analyzed, and new emerging themes were identified. This was

repeated for each participant until new themes were no longer emerging, thus signaling data saturation. In this study, data saturation was realized after the 15th participant..

## Data collection tool and procedure

A semi-structured interview guide (T_File) was used to collect data for the study. The guide was designed by the researchers based on the objectives of the study and was sectioned into four parts; A, B, C, and D. Ethical approval was received from the Ethics Review Committee of the Ghana Health Service before commencing data collection from participants. Also, written approval was obtained from the authorities of the Tamale Fistula Centre to access and recruit participants for the study. Participation was voluntary and informed consent of participants was obtained by their thumbprint or signing of consent forms after they were briefed about the study and allowed to ask questions and were satisfied with answers.

The data collection instrument was pre-tested with three women with obstetric fistula after which it was modified to include questions on general experiences of participants on obstetric fistula before narrowing the interview to the specific questions that pertained to the objectives of the study. Results of the pre-testing were, however, not included in the final data for analysis. COVID-19 protocols such as sanitizing, wearing a mask and physical distancing were adhered to during each interview session. An Audiotape recorder was used to capture all interviews along with taking field notes. Recruitment of participants and interviews by the first author began on 01/04/2021 and ended on 30/08/2021. Each participant had a separate interview session at different locations based on their choices and no other person was present during interviews. All interviews began with the introduction of the first author to the participant followed by general questions on the experience of the women with obstetric fistula before narrowing down to the specific questions in the study. Probing was used to further seek clarity of responses from participants where necessary. In all, 10 interviews were conducted in Dagbani, three (3) in English two (2) in Twi. Each interview session lasted a minimum of 35 minutes and a maximum of 50 minutes with none repeated during the study.

## Data analysis and management

The data was analyzed manually using Colaizzi's thematic analysis technique. The technique involves familiarization with data, identifying similar pattern, grouping common patterns into themes and providing in-depth description of themes. In applying the technique, the interviews conducted in Dagbani and Twi were translated into English by Dagbani and Twi language experts. The interviews were then transcribed verbatim by the first author. The first author read each transcript many times where meanings were formulated and categorized into themes and sub-themes. A concise description of the themes was finally written to represent each objective of the study. A total of five themes were deduced from the data, along with their respective sub-themes. Transcribed data was saved on a computer which was password-protected to prevent unauthorized access.

## Rigour

The trustworthiness of the study was achieved by using the framework of Guba and Lincoln [16] to ensure credibility, transferability, dependability, and confirmability. Credibility was achieved by ensuring that transcripts were interpreted by participants to confirm that they represented the information they provided during the interviews. Transferability was ensured by giving a detailed description of the study setting while dependability was achieved by subjecting the study to review by the second author who is an experienced researcher and supervisor. The data for the study was also made available for verification and certification by the second and third co-authors who supervised and reviewed the results. `

## Results

### Demographic information of participants

The demographic information of participants in the study was collected on age, educational background, occupation, marital status, number of children alive, and duration of the fistula. Out of the fifteen (15) participants in the study, the youngest participant was 20 years old while the oldest participant was 45 years old with one (1) unknown age but who affirmed to be older than 20 years. In terms of educational background, the majority of the participants (5, 33.3%) had no formal education, four (4, 26.6%) were primary school dropouts, three (3, 20%) were senior secondary school leavers and three (3, 20%) were junior secondary school leavers. Also, ten (10, 66.6%) participants were unemployed, four (4, 26.6%) were farmers with one (1, 0.06%) participant being a seamstress. Fourteen (14, 93.33%) of the participants were married while one (1, 6.67%) was a widow. About number of children, one (1, 6.67%) participant did not have a child while the rest of the participants (14, 93.33%) had between one (1) and five (5) children. Again, the highest duration of the fistula condition of participants was 84 months with the minimum duration being 3 months. The demographic information of participants is presented in Table 1.

### Themes and sub-themes

There were a total of five (5) main themes that emerged from the analysis of the data. These themes were coping strategies, justification for coping strategies, sources of support, and physical and psychological effects of coping strategies. In all, eight (8) sub-themes emerged under coping strategies, four (4) sub-themes under justification for coping strategies, three (3) sub-themes under sources of support, three (3) sub-themes under the physical effect of coping strategies, and three (3) sub-themes under the psychological effect of coping strategies. There was also an additional theme for experience with incontinence of urine and feces in which five (5) sub-themes emerged as presented in Table 2.

### Coping strategies

The coping strategies of women with obstetric fistula were identified and organized into eight (8) sub-themes for further analysis and better understanding of the data.

**Table 1. Demographic information of participants.**

| Pseudonym | Age (Years) | Educational background | Occupation | Marital status | Number of children alive | Duration lived with fistula (Months) |
|---|---|---|---|---|---|---|
| TF1 | 25 | Secondary level | Unemployed | Married | 1 | 3 |
| TF2 | 28 | Primary school drop out | Unemployed | Married | 2 | 4 |
| TF3 | 37 | None | Unemployed | Married | 5 | 12 |
| TF4 | 38 | Primary school drop out | Seamstress | Married | 2 | 36 |
| TF5 | 31 | Basic level | Farmer | Married | 3 | 48 |
| TF6 | 27 | Secondary level | Unemployed | Married | None | 13 |
| TF7 | 25 | Secondary level | Unemployed | Married | 1 | 6 |
| TF8 | 27 | Basic level | Unemployed | Married | 2 | 4 |
| TF9 | Unknown | None | Unemployed | Married | 2 | 24 |
| TF10 | 29 | Basic level | Farmer | Married | 2 | 12 |
| TF11 | 21 | None | Unemployed | Married | 3 | 60 |
| TF12 | 41 | None | Unemployed | Married | 3 | 48 |
| TF13 | 29 | Primary school drop out | Farmer | Married | 2 | 12 |
| TF14 | 45 | None | Unemployed | Married | 4 | 84 |
| TF15 | 20 | Primary school dropout | Farmer | Widow | 3 | 60 |

**Table 2. Themes and sub-themes.**

| Themes | Sub-themes |
|---|---|
| Coping strategies | • Using chamber pot chairs<br>• Wearing many clothes<br>• Use of sanitary products (old clothes, pampers and sanitary pads)<br>• Avoiding bulky foods<br>• Use of perfumes and detergents<br>• Frequent bathing and washing<br>• Characteristic slow walking style<br>• Medical urinary devices |
| Justification for coping strategies | • Proactive measure for incontinence<br>• Unusual noise control during walking<br>• Elimination of odour<br>• Prevention of splashing of feces |
| Support system of coping strategies | • Parents and siblings<br>• Health workers<br>• Neighbors |
| Physical effects of coping strategies | • Skin rashes<br>• Itching and pains |
| Psychological effects of coping strategies | • Fear<br>• Anxiety<br>• Grief |

The sub-themes were using chamber pot seats or chairs, wearing many clothes, use of rugs, pampers, or sanitary pads, avoiding bulky foods, use of perfumes or detergents, and frequently bathing and washing clothes. Other sub-themes under coping strategies were slow walking style and hiding in rooms.

***Using chamber pot chairs:*** Women in the study used chamber pots as seats at all times as a measure to cope with incontinence of feces or urine. The use of chamber pots as coping strategies was expressed as follows.

*"I used to sit on a chamber pot all the time so that in case the feces want to come it will just come out before I get up. I had to sit on the chamber pot from morning to evening for the period of 3 months before the reconstructive surgery".* ***(TF1)***

***Wearing of many clothes:*** Wearing multiple layers of clothes at a time was identified by participants as a coping strategy for obstetric fistula.

*"Before I was diagnosed by doctors, I was hearing strange noise from my private part after I delivered. People were always asking me for the reason behind that noise which I was too shy to tell them. For this reason, I try to wear several clothes to prevent the noise from getting louder so that people will not hear it".* ***(TF7***)

***Use of sanitary products (old clothes, diapers, and sanitary pads):*** The study also found that women with obstetric fistula used old clothes, diapers, and sanitary pads as a coping strategy to contain the incontinence of feces and urine.

*"I always use diapers and sometimes rags to make my sanitary pad for use. But when I use it for a long time, it affects me so I make sure that anytime it is soiled I will change it and bath immediately".* ***(TF4)***

***Avoiding bulky foods:*** The study also found that participants avoided eating bulky foods as a way to cope with obstetric fistula.

*"When I did the surgery, the doctor advised that I should be careful with the food I eat else the condition would come back. I was advised against pushing hard when I want to defecate so I should avoid constipation. I was asked to eat fruits and vegetables, foods that are high in protein and fiber to prevent constipation." (TF9)*

***Use of perfumes and detergents:*** The use of perfumes and high-scented detergents was also found in the study as a strategy adopted by the participants to cope with obstetric fistula.

*"My condition has made me smell of urine and feces. Even being myself, I don't like the way I smell let alone people around me. There is odor in my body and my clothes even in my room I feel the odor. So I used high-scented perfume and detergents for washing and cleaning."(TF12)* ***Frequent bathing and washing:*** The majority of the participants also cope with obstetric fistula by frequent bathing and washing of clothes.

*"Bathing and washing have been my strategy in dealing with my condition. Anytime I noticed urine or feces out of me I immediately took my bath and washed the clothes I used. If I don't do that I would not be comfortable at all" (TF11)*

***A characteristic slow walking style:*** Another coping strategy of women with obstetric fistula was the use of a slow walking style.

*"I walk carefully so that feces and urine would not easily flow and splash on my legs. I walk slowly to avoid splashing of fecal matter on my body". (TF14)*

***Medical urinary devices:*** Medical aids were found to be one of the coping strategies of participants in the study.

*"I was given a catheter which was connected to a bag at the hospital, and this collects the urine and feces without passing through my private parts. Whenever the bags get full, I will go back to the hospital; I usually go back to the hospital every week for the health professionals to change it. I am always indoors and only come out when I am going to the hospital". (TF6)*

### The justification for the use of the coping strategies

Three sub-themes emerged under this theme as follows;

**Proactive measure for incontinence**: This reason was concerned with participants' preparation in advance for incontinence of urine or feces.

*"I choose to sit on the chamber pot because I would not know when I feel to ease myself. So, I always sat on the chamber pot and when the feces came it would just flow into the chamber pot otherwise, I would soil myself." (TF1)*

Other participants who adopted the strategy of walking slowly indicated that it helped them to prevent the easy flow of feces and urine. The study found that these participants did not just walk slowly but they also kept their legs very close to each other during walking. All these were done to keep the feces from flowing out easily.

**Unusual noise control during walking:** Some participants in the study also indicated that they adopted their strategies to control noise they experienced when walking.

*"When my condition started, I used to hear a noise like 'kpakpara' anytime I was walking. This attracted people to stare at me a lot. So I decided to wear a lot of clothes anytime I am going out of my home just to help minimize the noise that comes out when I am walking so that other people will not hear it". (TF2).*

*Elimination of odor*: Getting rid of body odor was also found as a reason for the use of coping strategies among the participants.

*"My condition comes with more bad smelling so, my body, clothes, and my room would even smell bad. It is for this reason that I used high scented perfume and detergents for washing and cleaning". (TF12)*

## Sources of support for living with incontinence

In all, it was found that participants in the study had varying sources of support that contributed to their coping with the condition. While the main sources of their support were parents or siblings, health workers, and neighbors, the kind of support they received varied from participant to participant.

*Parents and siblings:* Parents and siblings of the women with obstetric fistula were found as sources where participants received support to cope with the condition.

*"As for my parents and siblings, their support for me was overwhelming. They cared for me before as well as stood for the surgery to be done and before that, they supported me with diapers when I was having incontinence of feces and urine till I left for my auntie. My support largely came from my parents during my difficult times." (TF8)*

*Health workers:* Participants in the study also disclosed that health workers in their communities served as a source of support for their coping strategies.

*"The community health nurses who come to do weighing in our community are aware of my condition and have been supporting me a lot. They bring me diapers, soap, and detergents anytime they come around." (TF10)*

*Neighbors and friends:* Participants also received support to cope with obstetric fistula from neighbors and friends.

*"I used to do washing for some UDS students and fetched water for them so they gave me more money. It was through their help that I was sent to the university health center and subsequently referred to the Tamale Teaching Hospital where I was diagnosed with Fistula. I was lucky I was close to some of the students; they bought me a lot of adult diapers, female boxer shorts, clothing, and toiletries like perfumes and soaps, Dettol." (FT14)*

## Physical effects of incontinence products

The use of various coping strategies by women with obstetric fistula was associated with physical effects. The physical effects identified by participants were skin rashes, itching, and pains.

*Skin rashes:* The majority of participants in the study identified skin rashes as one of the effects they suffered due to the strategies they adopted to cope with the incontinence of feces and urine.

*The adult diapers gave me rashes and the rashes are very painful and itching. I have to use a cream called 'Maame Dagomba' cream to treat it and it cost me financially." (TF14)*

*Itching and pains:* Women in the study also expressed how they suffered physical itching and pains due to the use of various strategies in coping with obstetric fistula.

*"My problem with the use of diapers or pampers is that it does not make me feel comfortable at all. I have observed that anytime I use them, I feel itching around my genitals and it is a painful experience." (TF11)*

**The psychological effect of coping strategies**

The study also found that women with obstetric fistula suffer psychological effects in various ways. Three (3) sub-themes that emerged were fear, anxiety, and grief.

*Fear:* Fear was found as one of the psychological effects experienced by participants in the study about the use of coping strategies.

*"Sometimes, when I pack my body with so many clothes before going out, it changes the way I look because my buttocks get increased and people outside would start asking why I look like that. I don't normally know what answer to give them but I hate such questions. Because of that I get afraid anytime I think of wearing so many clothes before going out." (TF7)*

Another cause of fear was found among participants who depended on the advice of health workers to eat only light foods. These women were found to be afraid that eating light foods only may cause them another sickness and death since they were not used to such foods only.

*"Even though I trust the doctors and nurses who advised me to take only light foods, I am also afraid of depending on only light foods. I am used to eating bulky foods because that gives energy but now they say I should eat only light foods so that I don't get constipation. I fear that I may die." (TF8)*

*Anxiety:* Women with obstetric fistula were also found to suffer anxiety in using coping strategies.

*"Because I don't control the incontinence of urine or toilet, I often have to wear diapers before walking to any place. The diapers are costly and I am always depending on other people's help to get some. When I realized my diaper was getting finished, I became so anxious because I didn't know how to go out without it. I can use my old clothes but it not able to absorb the flow like diapers. I think I am used to the diapers and that is why I get anxious when it is getting finished." (TF2)*

*Grief:* Grief was also found among women with obstetric fistula about their coping strategies. Some participants in the study were found to express their sorrows because they could not work or go out considering the strategies they adopted to cope with their situations. These were participants who made chamber pots their seats because they did not know when there would be a flow of feces and urine.

*"I feel so sad that I cannot work or go out without the chamber pot. I used to work and take care of my children but now I am always sitting on a chamber pot which I cannot stop using. How can I work in this situation? Ideally, chamber pots are not made for sitting but due to my unfortunate situation, I have to be using them as seats. The use of the chamber pot is making me sad." (TF7)*

*"The use of chamber pot is terrible and I am sad about that. If I will have another means to cope with my situation without having to sit on the chamber pot all day, I will be happy. I am not happy." (TF8)*

## Discussion

### Coping strategies of women with obstetric fistula

Various coping strategies by women with obstetric fistula have been identified in the literature in different parts of the world [14,15,17]. Obstetric fistula is generally associated with incontinence of urine or feces or both and for that reason greatly

affects the normal lives of victims. In most cases, women with obstetric fistula lose their sense of feeling to urinate or defecate and this consequently presents a situation where they need to prepare themselves in advance for the sudden flow of excreta. However, in the developed world where there is easy access to primary health care and high medical literacy among people, cases of obstetric fistula are easily detected and effectively treated through surgery leaving the women no need for adopting strategies to cope themselves. In developing countries like Ghana where access to integrated maternal health care is limited and there is a high rate of medical illiteracy, cases of obstetric fistula are high and take long time to be diagnosed hence leaving affected women limited opportunities to medically deal with the condition [18]. Besides, socio-cultural factors including stigma and traditional beliefs serve as barriers for women to seek medical attention. In other words, the fears of what society would say about the condition do not give women the confidence to openly seek healthcare. Similarly, the strong traditional beliefs in Ghanaian societies where medical conditions are attributed to spiritual cause adversary impact the health seeking behavior of women with obstetric fistula. These problems faced by the women in Ghana and other developing countries leaves them to adopt various coping strategies to deal with the condition [19]. However, the coping strategies adopted by women in this condition largely depend on the available resources at their disposal. While women with obstetric fistula who have resources may have effective ways of dealing with the condition, poor women affected by the condition may lack resources to effectively cope with the condition.

In the current study, various coping strategies were identified among women with obstetric fistula in the northern region of Ghana. These strategies have been classified into Physical-focused and Emotional-focused coping strategies.

## Physical-focused coping strategies

The study identified physical-focused coping strategies of participants as strategies used by women with obstetric fistula to maintain their physical well-being. As obstetric fistula is associated with incontinence of feces and urine, participants were found to adopt means that enabled them to reduce the physical effects of the condition on their lives. The study found that the physical-focused coping strategies of participants were frequent bathing and washing, use of perfumes and high-scented detergents, and characteristic 'slow walking style'.

Participants were found to bathe and wash their clothes frequently to keep themselves clean as much as possible. In addition to this, women in the study were found to use perfumes and high-scented detergents. Participants who used this form of strategies to cope with obstetric fistula were more focused on maintaining physical hygiene as the condition is associated with urine and feces. This finding supports the findings by Barageine [20] and Ahmed and Holtz [21] who concluded that the main coping strategies of women with obstetric fistula include frequent cleaning and bathing. The focus on hygiene by women in this condition is necessitated by the fact that the condition leaves them with odor in their bodies, clothes, and rooms. Notwithstanding, the fact of necessity, the strategy can be expensive considering the amount of money required by these women to buy perfumes and detergents such as soaps and other antiseptics. As a result, a sustainable source of support for women in this condition would contribute to ensuring effective physical hygiene for women with obstetric fistula.

Another finding of this study was the use of walking style to cope with obstetric fistula. Some women in the study adopted a 'slow walking style' as a strategy to cope with the soreness or pain caused by incontinence of feces and urine. This finding is not only new to findings in existing literature but calls for attention to explore more on how walking style could be used by women with obstetric fistula to cope with incontinence of feces and urine.

## Emotion-focused coping strategies

Emotion-focused coping strategies were identified in the study as those strategies adopted by women with obstetric fistula to keep them from stigma and embarrassment due to their condition. Participants in the study were found to use strategies such as sitting on a chamber pot, using diapers or sanitary pad and rugs, wearing many clothes. Participants who used chamber pots as seats to cope with their condition were women who lost the sense of feeling to urinate or defecate

and as a result, needed to always sit on chamber pots in readiness for flow of feces and urine. Even though this strategy may help in avoiding the possibility of participants soiling themselves with feces and urine unexpectedly; it also limits the women as they cannot move with the chamber pots everywhere and hence deprives them of work outside the home. Also, the use of diapers and sanitary pads to cope with obstetric fistula was found among participants in the study. While this coping strategy was expected among the majority of the women, participants in the study could not effectively adopt this strategy as they could not afford it and were largely dependent on donations from other people. This finding is consistent with the finding by Barageine [22], Jarvis [12], and Obed [18] who found that women with obstetric fistula were economically affected to the extent that they could not afford personal consumables such as soap, pomades, and sanitary pads for their personal up keeping. The finding suggests that there is a need for supporting systems to be instituted for women in such conditions until they are fully treated.

Participants in the study were also found to adopt wearing of many clothes as a coping strategy. This strategy was mostly used at the time participants wanted to go outside their homes for a gathering or to meet other people. The strategy was found among women who experienced noise during walking which they attributed to the effect of the obstetric fistula. While the participants believed the clothes helped them to reduce the noise and prevent other people from hearing it, the study could not establish the effectiveness of clothes as a means of controlling noise or the number of clothes they wore to achieve results. Again, granted that clothes are effective in reducing the noise, it means that poor women affected with the condition cannot use this means as a coping strategy as they may not have many clothes to wear. Further study is, therefore, necessary to determine more effective ways to contain the noise and the fecaluria discharges.

In all, the above coping strategies were positive emotional coping strategies by the Coping Circumplex Model. As proposed by Stanisławski [23], positive emotional coping involves the individual with a particular problem being kind and understanding to himself or herself while adopting ways to solve the problem regardless of whether those ways yield success or not. In the study, participants adopted ways to deal with a condition that they found as a problem that needed a solution irrespective of whether their strategies were successful or not.

## Justification for the use of coping strategies

The study found that coping strategies were used by the women for purposes of hygiene and for prevention of offensive odor that might lead to stigma and discrimination by other people.

As acknowledged in the study by Donnay [24] and Wall [25], the constant leaking of urine and feces due to obstetric fistula affects body parts like thighs and even the vulva. This is because; they are always wetted by feces and urine which result in odour. This seriously affects their hygiene and comfort. The majority of participants in the current study adopted coping strategies that enabled them to keep their hygiene. The use of chamber pots, diapers or sanitary pads, perfumes, and high-scented detergents by the women in the study was all toward ensuring their hygiene.

Another reason found among women in the study for the use of coping strategies was to prevent being suspected and stigmatized by other people. The study found that people discriminated against and stigmatized participants after knowing their conditions. This observation by the women supported the finding by Bashah [26] and Weston [27] which found that women with obstetric fistula are stigmatized and discriminated against due to leaking of urine, feces, and associated odor. To prevent stigmatization and discrimination by other people, the women used strategies like wearing several clothes before going out and sometimes hiding themselves to prevent other people from knowing that they were affected by such conditions. This negative societal attitude toward women with obstetric fistula compounds their pain rather than helping them deal with the condition. A major reason for this is wrong beliefs on the cause of the condition by society in general. This finding is important as it exposes the general lack of knowledge on obstetric fistula by most societies in Ghana. It is therefore imperative to educate the communities of these women against negative beliefs on obstetric fistula and call for support toward affected women.

## Sources of support for coping strategies of women with obstetric fistula (Emotional focused coping)

Participants in this study disclosed their sources of emotional support for the coping strategies as family, friends, neighbors, and health workers. This finding is similar to the finding of Sullivan, and O'Brien [28] that women with obstetric fistula perceived support from family members and partners. However, in this study, partners of women with obstetric fistula did not serve as a major source of support in their coping strategies.

In terms of family support, it was found that parents and siblings of participants offered them the moral support and cared for them before and during surgery and as well provided them with diapers until they were treated. The husbands and in-laws of participants however neglected the affected women. The fact of women with obstetric fistula being abandoned and neglected by their husbands as found in the study is an indication of ignorance on the part of men on the condition. Again, as argued by Khisa [29] the reason why women with obstetric fistula do not enjoy support from others is a result of their keeping secret their condition, the lack of support of partners of participants in this study can also be due to lack of communication between the women and their husbands. The inability of the women to share their situation with their husbands may leave the husbands in wrong suspicion leading to neglect of the women. Wrong beliefs on the cause of obstetric fistula may also be the reason for the neglect of affected women by their husbands. For instance, where delayed labor is believed to be caused by infidelity, the men are likely to blame the women for their conditions. Parents and siblings of participants in the study were found to support them with chamber pots and diapers to help the women cope with the condition. This finding means that women with obstetric fistula who have no parents or siblings may suffer the condition without any support.

The healthcare professionals and neighbours were supporting the women in terms of diapers, soap and detergents, clothing and other basic consumables. The findings from the study suggest that this source of support cannot be relied on by the affected women. The reason is that the support is subject to the kind of neighbors or healthcare workers the women may meet. For example, neighbors who appreciate the plight of women in this condition may out of sympathy contribute to help them cope with the condition. On the other hand, neighbors who may have wrong beliefs about obstetric fistula and think that the condition is a reward for the evil committed by the women may never empathize with them and hence would not offer any form of support to them. Similarly, healthcare professionals may be aware of the challenges of women with obstetric fistula due to their training and knowledge of the condition, but not all of them would have the sense of supporting these women out of their resources. The finding on the source of support in this study is important as it presents the need for an institutional form of support for women with obstetric fistula where they can effectively rely on it to cope until they are fully treated for their condition.

## Physical effects of physical-focused coping strategies on women with obstetric fistula

Considering the challenge of the condition on the body of affected women, most of the coping strategies were adopted to ensure personal hygiene. However, since some of the strategies lack medical basis and the fact that most women do not have adequate knowledge in ensuring good personal hygiene, their devised strategies sometimes affect them physically.

In this study, participants were found to suffer effects such as skin rashes, itching, and pains as a result of the coping strategies they used. This finding is different from that of Donnay [24], Khisa [19], and Wall [25], who found that obstetric fistula (not the coping strategies) destroys the vulva or thighs of women due to the incontinence of feces and urine. The effects found in this study were largely attributed to the use of diapers, sanitary pads, and self-made pads from old clothes. In normal circumstance, the use of diapers and sanitary pads do not naturally cause skin rashes. However, the finding in this study suggests that the effect of skin rashes suffered by participants may be due to improper use of diapers and sanitary pads. For instance, allowing the pads to stay too long on the body with feces or urine would lead to infections manifesting in the form of rashes on the body. This is the possible cause of the situation suffered by participants in the study as most of the women were poor and could not afford to change their diapers or sanitary pads frequently. This is

evident by the earlier finding that the diapers and sanitary pads used as coping strategies by the women were acquired as donations. As a result, it is likely that the participants as part of managing them for a long time, failed to change and replace as necessary hence resulting in infections. These findings call for the education of women with obstetric fistula on proper use of diapers and sanitary pads as well as coaching them to make their pads out of old clothes. This itching and pain suffered by participants in the study were the possible outcomes of the infections that caused the skin rashes. These physical effects can therefore be reduced or eliminated if women with obstetric fistula are provided with adequate diapers or sanitary pads and guided to use them properly.

## Psychological effects of coping strategies of women with obstetric fistula

Existing literature indicates that women with obstetric fistula suffer psychologically in different ways and the effects on them can be worse than the physical effects of the condition on them [9,25]. The findings of this study indicate that women with obstetric fistula suffer anxiety, grief, and fear as a result of the coping strategies they use following the effect of the condition on their lives.

The anxiety experienced by participants due to the use of coping strategies was largely found to be caused by the lack of financial capacity of the victims to afford items such as diapers and sanitary parts needed for coping with incontinence of feces and urine. This finding supports the finding of Barageine [20], Jarvis [12], and Obed [18] that women with obstetric fistula were impoverished and could not afford basic consumables. However, unlike the findings in existing literature that revealed only the economic problems faced by women with obstetric fistula, the finding in this study has gone further to establish that the affected women suffer anxiety because of their inability to personally afford these consumables that contribute to their coping strategies.

Another psychological effect of participants in the study was grief. Women with obstetric fistula in the study were found to suffer grief due to the effect of the coping strategies that had limited their ability to live normal lives. This was greatly expressed by participants who used chamber pot as a coping strategy and as a result, could not do normal activities they were engaged in before the condition. Also, study participants were found to suffer fear associated with the use of the coping strategies. These participants were concerned with the possibility of the coping strategies to expose them for people to know of their situation and hence discriminate or stigmatize them. This finding explains the need to intensify community awareness of obstetric fistula and campaign against the stigmatization of affected women. It is also a call for health practitioners to counsel victims to be bold and ignore negative public comments to avoid affected women from hiding themselves which contributes to delays in seeking medical care.

## Implications of the study findings

The study findings provide insight for healthcare providers to intensify awareness on the availability of medical treatment for obstetric fistula. In addition, it is a call for policy direction to support treatment of obstetric fistula so as to facilitate access to medical treatment among women affected by the condition.

## Limitations of the study

The study was conducted among women with obstetric fistula in the northern region and as a result, the findings can only speak to that setting though rigorous applicability principles of transferability were applied. Also, participants in the study were interviewed in the Dagbani language before translating into English for interpretation. Due to this, there is the possibility of misinterpretations or overstating the experiences because of translational errors. The inclusion of only women with obstetric fistula who were fluent in English, Dagbani or Twi could lead to selection bias of respondents. The findings of the study cannot be generalized due to the small sample size.

## Conclusions

Women with obstetric fistula in the Northern Region of Ghana adopt two main coping strategies: problem-focused where the victims were found to take action toward finding a solution to their problem and emotion-focused where they received support from various sources toward finding a solution to their problem. The main reasons for the use of coping strategies were to ensure personal hygiene and prevent being stigmatized by other people because of odour. In the use of these coping strategies, participants in the study were found to suffer physical effects such as skin rashes, itching, and pains. They also suffered psychological effects such as anxiety, fear, and grief due to the use of coping strategies. The causes of obstetric fistula are multiple, however the main predisposing factors in this study are prolonged or delayed labor and female genital mutilation. Many of the women are poor and deprived and have relationship glitches with their spouses. The women also suffered social humiliation and embarrassment. Hence the participant will benefit from related education on the management of obstetric fistula and general health literacy skills.

## Author contributions

**Conceptualization:** Patience Asupulie Akayila, Samuel Adjorlolo, Gideon Puplampu.

**Data curation:** Patience Asupulie Akayila.

**Formal analysis:** Patience Asupulie Akayila.

**Methodology:** Patience Asupulie Akayila, Samuel Adjorlolo, Gideon Puplampu.

**Resources:** Patience Asupulie Akayila.

**Supervision:** Samuel Adjorlolo, Gideon Puplampu.

**Validation:** Samuel Adjorlolo, Gideon Puplampu.

**Writing – original draft:** Patience Asupulie Akayila.

**Writing – review & editing:** Patience Asupulie Akayila, Samuel Adjorlolo, Gideon Puplampu.

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
