## [Decision Letter · Decision Letter 0]

13 Jan 2025

PMEN-D-24-00448

Coping strategies of women with obstetric fistula in the Northern Region of Ghana

PLOS Mental Health

Dear Dr. Akayila,

Thank you for submitting your manuscript to PLOS Mental Health. After careful consideration, we feel that it has merit but does not fully meet PLOS Mental Health’s publication criteria as it currently stands. Therefore, we invite you to submit a revised version of the manuscript that addresses the points raised during the review process.

We look forward to receiving your revised manuscript.

Kind regards,

Gellan Karamallah Ramadan Ahmed

Academic Editor

PLOS Mental Health

Journal Requirements:

1. Please provide an Author Summary. This should appear in your manuscript between the Abstract (if applicable) and the Introduction, and should be 150–200 words long. The aim should be to make your findings accessible to a wide audience that includes both scientists and non-scientists. Sample summaries can be found on our website under Submission Guidelines:

https://journals.plos.org/globalpublichealth/s/submission-guidelines#loc-parts-of-a-submission.

Additional Editor Comments (if provided):

Reviewers' comments:

Reviewer's Responses to Questions

**Comments to the Author**

1. Does this manuscript meet PLOS Mental Health’s publication criteria?

Reviewer #1: Yes

Reviewer #2: Yes

Reviewer #3: Yes

2. Has the statistical analysis been performed appropriately and rigorously?

Reviewer #1: N/A

Reviewer #2: N/A

Reviewer #3: Yes

3. Have the authors made all data underlying the findings in their manuscript fully available (please refer to the Data Availability Statement at the start of the manuscript PDF file)?

Reviewer #1: Yes

Reviewer #2: No

Reviewer #3: No

4. Is the manuscript presented in an intelligible fashion and written in standard English?

Reviewer #1: Yes

Reviewer #2: Yes

Reviewer #3: Yes

Reviewer #1: General comments and some key concerns:

1. It is an interesting study that is giving an insight on the ““Coping strategies of women with obstetric fistula in the Northern Region of Ghana”.

However, there some comments that needs to be addressed including:

• Authors institutions’ affiliation need to be more elaborated

•

• Long sentences and they need to be split up.

2. Abstract

• Aim or objective should be separated from the background and appear as sub-section

• The key findings from this study are lacking in the results section

• Similarly the conclusion is not based on the findings of the study

3. Introduction

• The term “Obstetric Fistula” need to be elaborated so that readers who are not health workers can understand what it means

4. Materials and methods section

• Line 84: should be Methodology instead of Materials and Methods

• Line 89: The statement – It was the largest……; Has it stopped to be the largest?

• What was the selection criteria of the study participants considering both inclusion and exclusion criteria!!

• How was this sample size of 15 women arrived at? And sample size seem to be small for the generalization of the findings

5. Results

• Table 1: Column of Duration of fistula should be “Duration of fistula (Months) and then remove the word month from the different patient values. And the duration of living with fistula by each participant, was it exactly whole numbers of months!!

• What were the outcomes of these coping mechanisms both positive and negative?

• Stress, depression, posttraumatic stress disorder, somatic complaints, and maladaptive coping are common problems among fistula patients, it is surprising that they were not mentioned or they were missed? See: these articles: doi: 10.1007/s12529-015-9466-2; DOI: 10.1007/s00384-015-2245-3 ; https://hamlinfistulauk.org/the-impact-of-fistula-on-mental-health/

6. Discussion

• What is the implication of the study findings?

• Among the limitations mentioned, I think small sample size can affect the generalization of the findings

7. Conclusion

• What were the main coping mechanisms that can be concluded from the findings?.

Reviewer #2: This manuscript addresses a significant public health issue by exploring coping strategies adopted by women suffering from obstetric fistula in the Northern Region of Ghana. The study is timely, well-conducted, and offers valuable insights into the experiences of a vulnerable population. The authors have employed a qualitative descriptive phenomenology approach, which is appropriate for capturing the lived experiences of participants. The findings are well-presented, with clear thematic organization and sufficient participant quotes to support the analysis. The research is particularly relevant given the ongoing challenges faced by women in resource-constrained settings, making it a valuable contribution to mental health literature by highlighting the psychosocial impacts of obstetric fistula.

The main claims of the paper are that women with obstetric fistula adopt both problem-focused and emotion-focused coping strategies to manage the physical and psychological burden of their condition, and that the lack of adequate resources and support exacerbates their mental health challenges. These claims are significant for the discipline of mental health as they underscore the complex interplay between physical health conditions and mental well-being.

Overall, the methodology is robust, and the use of thematic content analysis ensures systematic and thorough data interpretation. However, providing additional details on the process for determining data saturation with 15 participants would improve the methodological rigor and strengthen the credibility of the findings. Furthermore, expanding the discussion on cultural factors influencing coping strategies and healthcare-seeking behavior would provide a more nuanced interpretation of the results and deepen the reader’s understanding of the contextual challenges faced by the participants. The manuscript already mentions several cultural factors, such as stigma, traditional beliefs, and social expectations. However, expanding on how these cultural factors specifically influence the coping strategies of women with obstetric fistula and how they affect healthcare-seeking behavior would deepen the analysis.

Although the study identifies various support systems such as family, healthcare workers, and neighbors, a more detailed analysis of the nature, extent, and adequacy of these support mechanisms could enhance the findings. In particular, elaborating on the role of healthcare providers in aiding coping strategies would add depth to the discussion. Additionally, while the authors have indicated that data can be obtained upon request, it is advisable to deposit anonymized data in a publicly accessible repository. This step would enhance transparency and reproducibility, aligning the manuscript with PLOS Mental Health’s data-sharing policies.

In conclusion, this manuscript makes an important contribution to understanding how women with obstetric fistula cope in resource-constrained settings. With minor revisions to address the areas of data saturation, cultural context, support systems, and data availability, the manuscript will be well-suited for publication in PLOS Mental Health.

Reviewer #3: I thoroughly enjoyed reading this manuscript and it addresses a timely and relevant topic, making a significant contribution to the field and well-being of women in developing countries. The findings have significant implications for both theory and practice, offering valuable recommendations. The study’s results are impactful and have the potential to influence future research and policy in the region.

Specific comments.

Line 17 - colon and semi colon together after incontinence

Line 24 - Northern Region (More likely for readers to ask... of where?) Might be useful to emphasise that the area is called that or bracket Ghana. Also Line 507 - Capital first letters?

Line 35 - Does this constitute body odour/ smells or anything physical that announces their condition? Useful to mention this as reader will need to know in lines 60-62 why employers won't keep them. In more developed countries, incontinence pads are used so readers in such areas need to understand the extent e.g. of heat... (first mentioned in Table 2 - Line 172)

Line 40 - space between as and being

Line 41, 56 and throughout - Check citation dates - APA

Line 52 - please reword

Line 102 - Please reword - Were the women aged 18 years or more or diagnosed for 18 years or more? (confirmed in line 160 and Table 1)

Line 130 - might be useful to summarise what the Colaizzi method is.

Line 152 - repeated phrase - years old; also, can assurance be given that the unknown age individual was not under 18 years old?

Line 156 - reword - four were farmers with one participant was...

Line 157-missing comma

Line 172 - Table 1 - Itching and pains

Line 192 - Reconsider use of pampers except participants meant the specific brand - consider diapers or nappies?

Line 329 - reconsider - take longer time

Line 520 - multiple, however.

Line 526 - "The PLOS Data policy requires authors to make all data underlying the findings described in their manuscript fully available without restriction, with rare exception."

Please add in dates with the citations, APA (Line 41) please revise reference citation to be as PLOS mental health guideline..

**Do you want your identity to be public for this peer review?** For information about this choice, including consent withdrawal, please see our Privacy Policy

Reviewer #1: No

Reviewer #2: No

Reviewer #3: **Yes: ** Tola Awe

---

## [Decision Letter · Decision Letter 1]

10 Apr 2025

PMEN-D-24-00448R1

Coping strategies of women with obstetric fistula in the Northern Region of Ghana

PLOS Mental Health

Dear Dr. Akayila,

Thank you for submitting your manuscript to PLOS Mental Health. After careful consideration, we feel that it has merit but does not fully meet PLOS Mental Health’s publication criteria as it currently stands. Therefore, we invite you to submit a revised version of the manuscript that addresses the points raised during the review process.

We look forward to receiving your revised manuscript.

Kind regards,

Gellan Karamallah Ramadan Ahmed

Academic Editor

PLOS Mental Health

Journal Requirements:

Additional Editor Comments (if provided):

Reviewers' comments:

Reviewer's Responses to Questions

**Comments to the Author**

Reviewer #1: All comments have been addressed

Reviewer #2: All comments have been addressed

Reviewer #3: All comments have been addressed

publication criteria?

Reviewer #1: Yes

Reviewer #2: Yes

Reviewer #3: Yes

3. Has the statistical analysis been performed appropriately and rigorously?

Reviewer #1: N/A

Reviewer #2: N/A

Reviewer #3: N/A

4. Have the authors made all data underlying the findings in their manuscript fully available (please refer to the Data Availability Statement at the start of the manuscript PDF file)?

Reviewer #1: Yes

Reviewer #2: Yes

Reviewer #3: Yes

5. Is the manuscript presented in an intelligible fashion and written in standard English?

Reviewer #1: Yes

Reviewer #2: Yes

Reviewer #3: Yes

Reviewer #1: General comments and some key concerns:

1. Thanks authors for addressing issues raised in the manuscript entitled “Coping strategies of women with obstetric fistula in the Northern Region of Ghana” as a way to improve its quality. However, there are still some issues that need to be addressed to further improve the paper as below:

• Authors institutions’ affiliation need to be more elaborated i.e. institution, city and country and 2 and 3 are not clear in the institution affiliations

2. Abstract

• Objective should to be “study assessed the Coping strategies of women with obstetric fistula in the Northern Region of Ghana”

• Conclusion is not based on the findings of the study and therefore it should be re-written.

• Similarly the conclusion is not based on the findings of the study

• What is this author summary? And it should be incorporated in the Introduction sub-section

3. Introduction

• Line 56-59: As authors try to define the term “Obstetric Fistula”; they should avoid such as “In simple words….”

• Is there any literature available to support some of the coping mechanisms such women have developed?

•

4. Methodology

• Line 114: The statement – It was the largest……; Has it stopped to be the largest?. This comment was raised but the authors have not tried to attempt to address it

• Line 127: Women who were not fluent in English, Dagbani, or Twi languages, they were excluded. And therefore if many of them were not fluent in those languages, they would not participate in the study. I don’t think this is enough to exclude them from the study thus creating selection bias. Le the author think through this exclusion criteria

• Line 130: the statement “…traced were excluded in the study….”, how can one exclude what doesn’t exist?

• Who collected the data or administered the tool to the study participants?

5. Results

• Table 1: what is the unit of age?, Similarly, Duration of fistula (Months) should be “Duration lived with fistula (months)”

• In table 2: “Psychological effects of using incontinence products” , this statement is not clear. These individuals already experience incontinence, and so how can they go ahead and use incontinence products?

• These sub-themes in table 2, I suggest that they should be treatment separately especially those which are not similar. Otherwise the table is confusing on which is which?

6. Discussion

• Line 522: The statement “Psychological side effects of coping strategies on women with obstetric fistula”, if these are coping mechanisms, how then can they cause psychological effects!!!

• Line 552: Limitations of the study, I think the small sample size can affect generalizability of the conclusion, any comment on this?

7. Conclusion

• What were the main coping mechanisms that were found in the study such that a conclusion can be made on them? See the title and objectives of the study.

Reviewer #2: Thank you for the opportunity to review this revised manuscript. I appreciate the authors’ thoughtful responses to the initial feedback and the care taken in improving the clarity and depth of the work.

This version shows clear responsiveness to earlier comments, with improved methodological clarity, a deeper exploration of cultural and mental health dimensions, and enhanced data transparency. It offers a deeper understanding of the coping strategies used by women living with obstetric fistula, framed within a richer cultural context. The added explanation of data saturation and the expanded discussion of support systems are especially helpful, and the updated data availability statement improves the study’s transparency and alignment with PLOS policies.

While a minor language edit could further improve clarity and flow, the manuscript is well-structured and will be a strong contribution to PLOS Mental Health.

Reviewer #3: This is very interesting and relevant piece of work. A brief definition of obstetric fistula can be added to the abstract to help readers understand the condition before reading further.

**Do you want your identity to be public for this peer review?** For information about this choice, including consent withdrawal, please see our Privacy Policy

Reviewer #1: No

Reviewer #2: No

Reviewer #3: **Yes: ** Tola Awe

---

## [Decision Letter · Decision Letter 2]

19 May 2025

PMEN-D-24-00448R2

Coping strategies of women with obstetric fistula in the Northern Region of Ghana

PLOS Mental Health

Dear Dr. Akayila,

Thank you for submitting your manuscript to PLOS Mental Health. After careful consideration, we feel that it has merit but does not fully meet PLOS Mental Health’s publication criteria as it currently stands. Therefore, we invite you to submit a revised version of the manuscript that addresses the points raised during the review process.

We look forward to receiving your revised manuscript.

Kind regards,

Gellan Karamallah Ramadan Ahmed

Academic Editor

PLOS Mental Health

Journal Requirements:

Additional Editor Comments (if provided):

Reviewers' comments:

Reviewer's Responses to Questions

**Comments to the Author**

Reviewer #1: All comments have been addressed

publication criteria?

Reviewer #1: Yes

3. Has the statistical analysis been performed appropriately and rigorously?

Reviewer #1: Yes

4. Have the authors made all data underlying the findings in their manuscript fully available (please refer to the Data Availability Statement at the start of the manuscript PDF file)?

Reviewer #1: Yes

5. Is the manuscript presented in an intelligible fashion and written in standard English?

Reviewer #1: Yes

Reviewer #1: General comments and some key concerns:

Dear authors, thank you for making the revision to the manuscript and responding to the comments on the manuscript titled “Coping strategies of women with obstetric fistula in the Northern Region of Ghana”. However, the following are some few issues highlighted that need further attention.

• There are still some grammar issues and the use of tenses that still need to be improved i.e. line 258, 323, 557 etc. The authors should go through the whole document and correct all the grammar issues.

• Authors’ institutions’ affiliation need to be more elaborated and author number 2 has no institution. The authors have not responded to this comment.

1. Abstract

Line 16-17: The statement “Participation was by written informed consent and interviews were recorded ……”is not clear and it needs to be paraphrased.

2. Introduction

Line 38-42: The authors should merge the two sentences since they look similar. In line 49, the authors are mentioning that there many factors responsible for obstetric fistula, can the authors list them.

3. Results

The authors should have captured the associated risk factors to obstetric fistula by the study participants to cater for patient TF2. Patient TF2 in table 1, according to the definition given on obstetric fistula, the patient has no child and this assumes she has never given birth, can the authors explain why she had obstetric fistula? And this should be discussed in the discussion section.

4. Discussion

The authors should discuss the associated risk factors to obstetric fistula and then cater for patient TF2.

**Do you want your identity to be public for this peer review?** For information about this choice, including consent withdrawal, please see our Privacy Policy

Reviewer #1: No

---

## [Decision Letter · Decision Letter 3]

6 Jul 2025

PMEN-D-24-00448R3

Coping strategies of women with obstetric fistula in the Northern Region of Ghana

PLOS Mental Health

Dear Dr. Akayila,

Thank you for submitting your manuscript to PLOS Mental Health. After careful consideration, we feel that it has merit but does not fully meet PLOS Mental Health’s publication criteria as it currently stands. Therefore, we invite you to submit a revised version of the manuscript that addresses the points raised during the review process.

We look forward to receiving your revised manuscript.

Kind regards,

Gellan Karamallah Ramadan Ahmed

Academic Editor

PLOS Mental Health

Journal Requirements:

Additional Editor Comments (if provided):

Reviewers' comments:

Reviewer's Responses to Questions

**Comments to the Author**

Reviewer #1: All comments have been addressed

publication criteria?

Reviewer #1: Yes

3. Has the statistical analysis been performed appropriately and rigorously?

Reviewer #1: Yes

4. Have the authors made all data underlying the findings in their manuscript fully available (please refer to the Data Availability Statement at the start of the manuscript PDF file)?

Reviewer #1: Yes

5. Is the manuscript presented in an intelligible fashion and written in standard English?

Reviewer #1: Yes

Reviewer #1: Minor comments:

Dear authors, thank you for making the revision to the manuscript and responding to the comments on the manuscript titled “Coping strategies of women with obstetric fistula in the Northern Region of Ghana”.

However, the following minor issues highlighted need attention.

1. Results

The authors should address the issue of Patient TF6 in table 1, according to the definition given on obstetric fistula, the patient had no children and this assumes she has never given birth, can the authors explain why she had obstetric fistula? And this should be discussed in the discussion section.

**Do you want your identity to be public for this peer review?** For information about this choice, including consent withdrawal, please see our Privacy Policy

Reviewer #1: No

---

## [Decision Letter · Decision Letter 4]

7 Aug 2025

Coping strategies of women with obstetric fistula in the Northern Region of Ghana

PMEN-D-24-00448R4

Dear Ms Akayila,

We are pleased to inform you that your manuscript 'Coping strategies of women with obstetric fistula in the Northern Region of Ghana' has been provisionally accepted for publication in PLOS Mental Health.

Best regards,

Gellan Karamallah Ramadan Ahmed

Academic Editor

PLOS Mental Health

Reviewer Comments (if any, and for reference):

Reviewer's Responses to Questions

**Comments to the Author**

Reviewer #1: All comments have been addressed

publication criteria?

Reviewer #1: Yes

3. Has the statistical analysis been performed appropriately and rigorously?

Reviewer #1: Yes

4. Have the authors made all data underlying the findings in their manuscript fully available (please refer to the Data Availability Statement at the start of the manuscript PDF file)?

Reviewer #1: Yes

5. Is the manuscript presented in an intelligible fashion and written in standard English?

Reviewer #1: Yes

Reviewer #1: Dear authors, thanks and all the concerns have been addressed.

**Do you want your identity to be public for this peer review?** For information about this choice, including consent withdrawal, please see our Privacy Policy

Reviewer #1: No
